# Experimental Investigation on Axial Compression of Resilient Nail-Cross-Laminated Timber Panels

Moncef L. Nehdi [1],*, Yannian Zhang [2], Xiaohan Gao [2], Lei V. Zhang [1] and Ahmed R. Suleiman [1]

[1] Department of Civil and Environmental Engineering, Western University, London, ON N6G 1G8, Canada; lzhan666@uwo.ca (L.V.Z.); asuleim3@uwo.ca (A.R.S.)

[2] School of Civil Engineering, Shenyang Jianzhu University, Liaoning 110168, China; zyntiger@163.com (Y.Z.); gxhjl1996@163.com (X.G.)

* Correspondence: mnehdi@uwo.ca; Tel.: +1-(519)-661-2111 (ext. 88308)

**Abstract:** Conventional cross-laminated timber is an engineered wood product consisting of solid-sawn lumber panels glued together. In this study, the structural behavior of solid wood panels of Nail-Cross-Laminated Timber (NCLT) panels connected with nails instead of glue was studied. The failure mode and nail deformation of the novel NCLT panels under axial compression load using eight full-scale NCLT panels was investigated. The effects of four key design parameters, namely, the nail type, number of nails, nail orientation angle, and nail slenderness ratio on axial compression performance of NCLT panels were also analyzed. In addition, a formula for predicting the axial compression bearing capacity of NCLT panels was developed. For calculation of the slenderness ratio, the moment of inertia of the full section or the effective section was determined based on the nail type, number of nails, angle of nail orientation and number of layers of the plate. Results showed that specimens connected by tapping screws had best compressive performance.

**Keywords:** NCLT panels; timber; nail; axial compression; nail slenderness; nail orientation

## 1. Introduction

In the 1970s, the concept of cross-laminated timber (CLT) was first proposed in Europe [1]. CLT, commonly known as glulam, is a wood-panel product made by gluing layers of solid-sawn timber together. In lumbers, high tensile strength is achieved along the grain, and high compressive strength is attained along the horizontal grain. Therefore, to provide desirable tow-way mechanical properties, each layer of timber is orientated perpendicular to adjacent layers and glued on the wide faces of each timber, usually in a symmetric way so that the outer layers have the same orientation [2]. CLT panels have various advantages, including excellent mechanical properties [3], sound insulation [4], energy-saving potential [5], fire resistance [6,7], high durability [8], and construction convenience, owing to prefabrication [9]. CLT panels were used in many mid-rise condominiums for their preferable characteristics such as prefabrication and lightweight nature compared to other construction materials [10]. With the rapid development of CLT over the past decades, CLT panels have become gradually favored for their superior performance in bridges, timber decks, landscaped gardens, and other applications.

In recent years, substantial research on CLT panels has been conducted, significantly promoting the development of glued wood structures [11]. For example, Gavric et al. [12] conducted an experimental study in 2010 on the anchor connection between two adjacent CLT wall panels. Their results showed that the arrangement of the joints and the depth of the nail threads directly affected the deformation of the entire test specimens. Amini et al. [13] systematically investigated the effects of several parameters by conducting quasi-static cyclic tests on CLT shear wall systems. It was reported that panels with a higher aspect ratio (4:1) exhibited weaker stiffness and much higher deformation capacity when compared to moderate aspect ratio panels (2:1). In another study by Orlowski [14],

a novel prefabricated stiffened engineered timber wall was proposed. An axial load test was performed to examine the performance of the wall system under different design scenarios. It was found that the change of loading condition resulted in a steep decline in eccentricity. On the basis of this study, Orlowski and Kennedy [15] proposed a novel analytical approach via mathematical modelling to understand and predict the behaviour of the wall system.

While CLT panels have numerous benefits, making them attractive as a building material [16–18], there are still some problems to be solved for CLT panels to gain wider acceptance [19,20], including (1) relatively low service life of the adhesive connection [21]; (2) environmental pollution related to using inferior glues [22]; and (3) poor connection exhibiting glue joint cracking and plate layer separation [23]. Therefore, the increasing concern for the safety and long-term performance of engineered wood products calls for the development of more durable laminated timber.

Nail-laminated timber (NLT) is a solid structural lumber element mechanically laminated using nails. NLT can be produced with any dimension. In addition, NLT panels can be easily prefabricated and installed at construction sites. Compared to conventional glulam, nail connections in NLT have the advantages of a tight enclosure, are simple to manufacture, and have enhanced safety and reliability. Considerable research efforts have been invested in a wide range of research on NLT [24–26]. For example, Derikvand et al. [27] investigated the bending performance of NLT constructed of fast-grown plantations of eucalypt by conducting vibration and four-point bending tests. The results showed that the bending performance of the NLT panels made of two species of the eucalypt surpassed those of commercial NLT products in the previous literature. Similarly, the flexural performance of NLT crane mats was experimentally examined by Herberg [28], and the results were compared with current design recommendations for structural applications. It was concluded that the measured flexural strength of NLT specimens using a random lamination layup design was about 1.5 times higher than that of symmetric layup design. In another study, the structural performance of timber-concrete composite (TCC) systems consisting of an NLT panel connected to a concrete slab via shear connections was examined by Hong using impact loading and a quasi-static monotonic loading test [29]. It was found that this nail-laminated TCC system exhibited slightly higher elastic bending stiffness than the theoretic prediction.

Recently, nail-cross-laminated timber (NCLT) panels combining characteristics of CLT and NLT have been created. Like CLT, each layer of NCLT boards is placed in an orthogonally alternating orientation to the adjacent layers, and laminations are jointed using nails rather than glue. Despite the wealth of knowledge regarding the performance of CLT panels, very few studies have addressed the engineering properties of NCLT. Zhang et al. [30] investigated the flexural performance of NCLT panels by exploring the effect of various parameters, including nail incidence angle, nail type, the total number of nails, and the number of layers. Results indicated that NCLT panels exhibited better flexural performance when compared to conventional CLT panels. In a similar study, the bending performance and rolling shear of NCLT panels compared to conventional NLT and CLT panels were investigated by Hosseinzadeh et al. [9]. It was observed that the bending strength in the three types of panels did not show significant differences. In addition, the rolling shear failure in NCLT was more ductile compared to that of CLT.

While there has been extensive research on CLT and NLT, the current understanding of new NCLT remains limited. Therefore, in this study, drawing tests on five groups of nailing plates and axial compression tests on eight full-scale NCLT panels were conducted to investigate the anchoring effect of nails and the axial compression performance of NCLT panels. Design parameters of the drawing test include the nail type, number of nails, and nail orientation angle. The main design parameters of the axial compression test include the nail type, nail angle, nail slenderness ratio, and the number of nails in the overlapping area. Theoretical developments are presented for developing a predictive model for the experimental behavior to aid engineers in effective NCTL design.

## 2. Materials and Methods

### 2.1. Test Specimens

A total of 8 full-scale NCLT panels were designed for axial compression testing. The test specimens were made of Canadian spruce-pine-fir woods (Grade III) cut along the path of the fiber growth, with dimensions (length × width × thickness) of 3000 mm × 184 mm × 38 mm. Round nails, threaded nails, and self-tapping screws made of stainless steel were selected to construct the test specimens. And the original woods were then processed into vertical-grained boards and horizontal-grained boards, with specifications of 2220 mm × 184 mm × 38 mm and 370 mm × 184 mm × 38 mm, respectively. The light round nails were nailed using a steel hammer; the screw nails were nailed using a nail gun, and the self-tapping screws were driven by an electric drill. The nail arrangement meets the requirements of minimum margin, end distance, and intermediate distance. The thickness of the nail into the layer satisfies the minimum thickness requirement of the wood component in nail connections, as shown in Table 1. The test specimen parameters are presented in Table 2. The fabrication of NCLT panels is depicted in Figure 1.

**Table 1.** The minimum distance of nail arrangement.

| *a* | Rift Grain | | Band | | |
|---|---|---|---|---|---|
| | **Mid-Range** | **Terminal Distance** | **Mid-Range $S_2$** | | **Margin** |
| | $S_1$ | $S_0$ | **Homogeneous Column** | **Staggered or Slanted Column** | $S_3$ |
| $a \geq 10\,d$ | $15\,d$ | | | | |
| $10\,d > a > 4\,d$ | Fetch insertion value | $15\,d$ | $4\,d$ | $3\,d$ | $4\,d$ |
| $a = 4\,d$ | $25\,d$ | | | | |

Note: *d* is the diameter of the pin; a is the thickness of the member being pierced.

**Table 2.** Specimen Parameters.

| Specimen Number | Nail Type | Number of Plies | Nail Number | Nail Angle | Nail Length /mm | Nail Diameter/ mm | Sectional Dimension/ mm² | Specimen Height/ mm |
|---|---|---|---|---|---|---|---|---|
| ZYG54-0-70 | Round nail | 5 | 4 | 0 | 70 | 3.5 | 370 × 190 | 2220 |
| ZYL54-0-70 | Screw nail | 5 | 4 | 0 | 70 | 2 | 370 × 190 | 2220 |
| ZYZ54-0-70 | Self-tapping screw | 5 | 4 | 0 | 70 | 3 | 370 × 190 | 2220 |
| ZYL34-0-70 | Screw nail | 5 | 4 | 0 | 70 | 2 | 370 × 114 | 2220 |
| ZYL74-0-70 | Screw nail | 5 | 4 | 0 | 70 | 2 | 370 × 266 | 2220 |
| ZYL58-0-70 | Screw nail | 5 | 8 | 0 | 70 | 2 | 370 × 190 | 2220 |
| ZYG54-30-80 | Round nail | 5 | 4 | 30 | 80 | 3.5 | 370 × 190 | 2220 |
| DZYL54-0-70 | Screw nail | 5 | 4 | 0 | 70 | 2 | 370 × 190 | 1800 |

Note: the principle of specimen numbering: first, two letters "ZY" represent axial compression; the third letter represents nail type; the fourth letter represents the number of laminate layers of the specimen; the fifth letter represents the number of nails in the superimposed area; The sixth letter stands for the angle of the nail; the seventh letter stands for the length of the nail, "D" indicates deep specimen with increased height.

### 2.2. Test Setup, Iinstrumentation and Experimental Procedures

All axial compression tests on NCLT panels were carried out using a 5000 kN hydraulic testing machine in the Structural Engineering Laboratory of Shenyang Jianzhu University. Protective steel sleeves were placed at the and bottom of all test specimens. In order to ensure that both ends of the test specimens could rotate freely, a ball-hinge support was used at both ends of the test specimens for loading. The test setup for axial compression loading is shown in Figure 2.

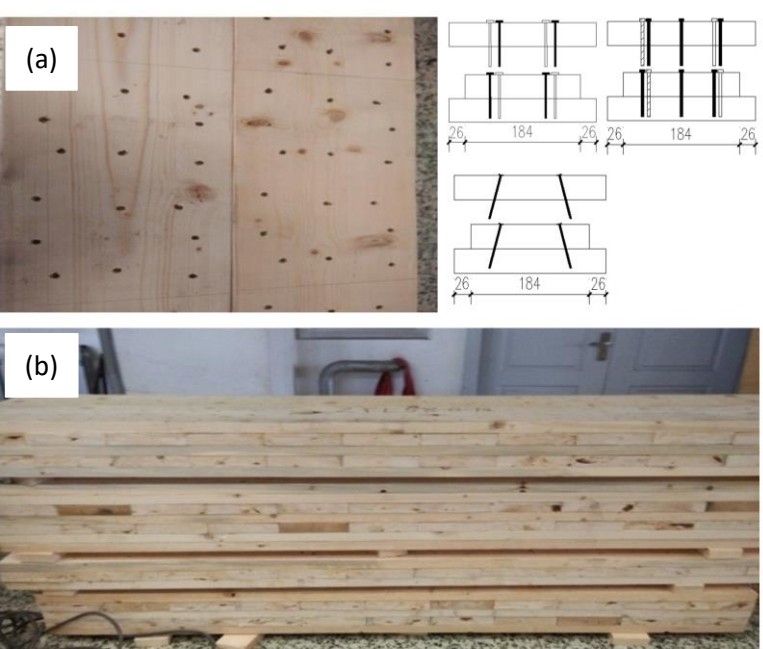

**Figure 1.** Fabrication of NCLT panels. (**a**) Arrangement and angels of nails; and (**b**) NCLT panels ready to be tested.

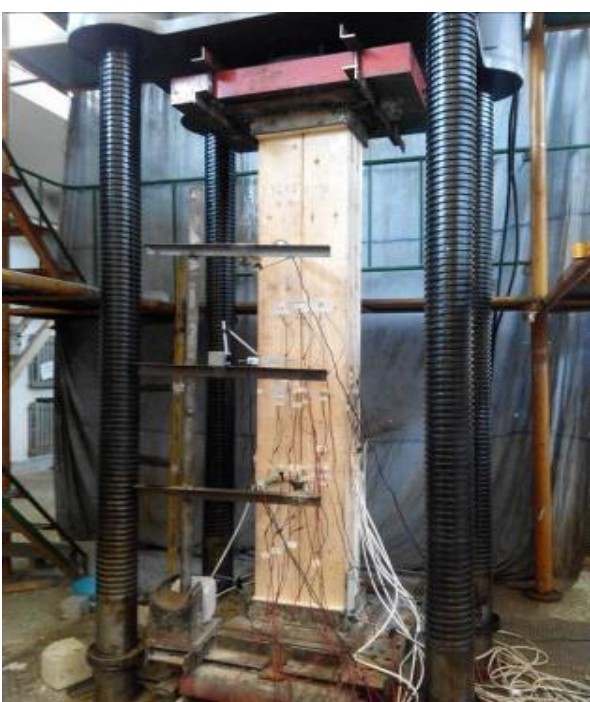

**Figure 2.** Test setup for axial compression load.

The main components of test measurement include the bearing capacity of the test specimens (including the yield load and ultimate load), the displacement at four points of the test specimens, the strain of the lamellar layer, and the development and distribution of cracks. Figure 3 shows the placement of strain gauges. Three displacement gauges were arranged along the height of the four points to measure the deflection of the specimen under load. The displacement gauges were numbered from top to bottom as W1, W2, W3, and the span-middle displacement meter range was 200 mm, while the upper and lower two displacement meter ranges were 100 mm. Lamellar strains were measured using strain gauges, with a gauge of 100 mm, a resistance of 120 ± 0.2 Ω, a sensitivity

coefficient of 2.097 ± 0.10%, and a gate length × gate width of 100 mm × 3 mm. The specific arrangement and positiond of the strain gauges are depicted in Figure 3. Strain gauges were attached to the center of the layer height, and at one-third of the two sides on both sides of the test specimens with and without nails. Four strain gauges were arranged at the width of the center and middle plate gap of two vertical grain solid wood plies. Strain gauges of the nail side were numbered S1 to S12, while strain gauges of the side without nails were numbered 1 to 12, respectively. Strain gauges were also arranged at the center of the board height on the testing panel's left and right sides. To mitigate the gap between the two horizontal-grained solid wood boards, strain gauges were placed at the center of the horizontal-grained board on the side of the board seam. Strain gauges were listed as C1 to C3 on the left side, and the gauges named D1 to D3 were installed on the right side.

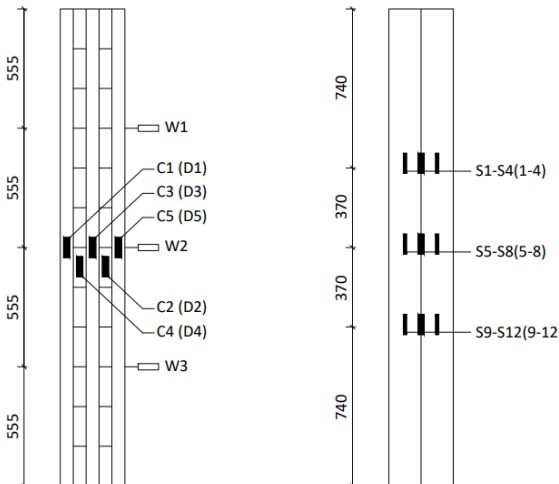

**Figure 3.** Measuring points arrangement.

Strain gauges were also arranged at the center of the plate height on both sides of the test panel. To avoid opening the seams of the two horizontally grained solid wood layers, strain gauges were arranged at the center of the horizontally grained solid wood layers of the plate gap. The development and distribution of cracks were monitored using a magnifying glass, a steel ruler, and a vernier caliper. The cracking load was recorded, and the development position of the splitting was marked. The maximum crack width and length at the time of failure were also measured.

### 2.3. Material Properties

The wood used in making specimens was Canadian spruce-pine-fir, referred to as SPF specification material, with dimensions (length × width × thickness) of 3000 mm × 184 mm × 38 mm. It was processed into vertical solid wood board and transverse solid wood board layers with specifications of (2220 × 184 × 38) mm and (370 × 184 × 38) mm, respectively. The physical performance indicators of the wood material are shown in Table 3.

**Table 3.** Physical properties of SPF.

| Varieties of Trees | Bending Strength/MPa | Along-Line Compression Strength/MPa | Transverse Compression Strength/MPa | Density/kg/m³ | Rate of Water Content |
|---|---|---|---|---|---|
| Spruce | 69.5 | 37.8 | 4.1 | 430 | 12% |
| Pine | 68.9 | 40.8 | 3.6 | 445 | 12% |
| fir | 69.1 | 38.9 | 3.8 | 440 | 12% |

## 3. Experimental Results and Analysis

### 3.1. General Behavior

At the start of the preloading of test specimens, a continuous subtle sound appeared, likely originating from tightening the support in close contact with the test specimen. Early in the loading process, all test specimens showed flexural deformation with small lateral displacement. As the load increased, thin cracks first appeared in the wood fibers at the joints. When loading reached about 80% of the ultimate load, the specimen began to exhibit lateral deformation, and the crack near the knot continued to extend to the decayed area with the intermittent sound of wood fiber squeezing. Creases appeared on the pressed side of the wood fibers. When approaching the ultimate load, the deformation of the specimen accelerated, and continuous internal sound could be heard. In addition, some fibers near the knuckle of the tension side and the decayed part started breaking, and cracks between the knots and the decayed parts spread along the direction of the lamina. When the load reached to about 80% of the ultimate load, the deformation became much more obvious. The solid wood panels with vertical grain on the tension side were broken along the entire section where knots and decayed parts were located. Plate layer bending and local bulging occurred, and cracks between the wood section and decadent part continued to increase along the grain direction of the plate layer up to 92 mm. There was only a slight crack in the center of the height of the plate on the tension side, which was hardly distinguished by the naked eye. The specimen damage pattern is shown in Figure 4.

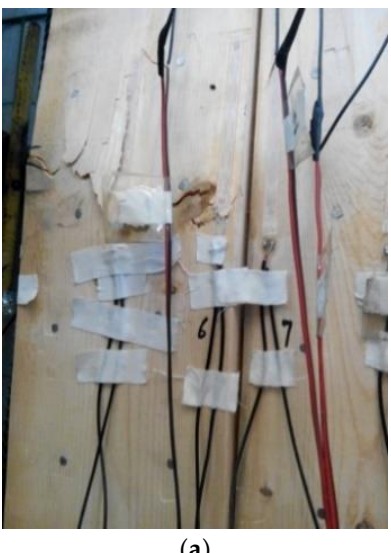 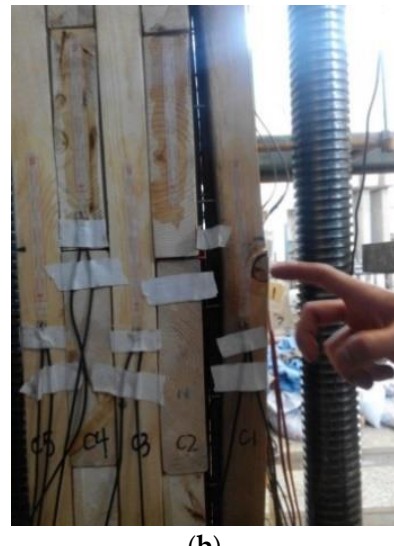

(**a**)                                                     (**b**)

**Figure 4.** Tension bending failure. (**a**) Test specimens ZYG54-0-70 bent; (**b**) Local drum bending of specimen ZYG54-0-70.

### 3.2. Failure Mode

Under the action of the applied force on 25 test specimens, damage patterns could be classified into three main forms, namely, nails pulling-out, wood board splitting along the outer surface of the nail, and their combination. Notably, the nails of all the test specimens were not broken. When reaching the ultimate load, the test specimens began to lose load bearing capacity, initiating varying degrees of damage. The variation coefficient of the ultimate drawing capacity of five specimens in each specimen group was between 4.2% and 8.9%. Therefore, the average value of five specimens in each group was taken as the ultimate drawing load of the test group, and the main test results are reported in Table 4.

**Table 4.** Main results.

| Specimen Group | Nail Type | Number of Nails | Nail Angle | Ultimate Pull-Out Load/kN |
|---|---|---|---|---|
| LG4-0 | Round nail | 4 | 0° | 0.37 |
| LG4-30 | Round nail | 4 | 30° | 1.11 |
| LL4-0 | Screw nail | 4 | 0° | 1.40 |
| LL8-0 | Screw nail | 8 | 0° | 3.29 |
| LZ4-0 | Self-tapping screw | 4 | 0° | 4.09 |

"L" represents the pull-out test; the second letter represents the nail type, where "G" indicates round nail, "L" indicates threaded nail, and "Z" indicates self-tapping screw; the third digit represents the number of nails in the overlap area; the fourth/fifth digit represents the nail angle.

Under axial pressure, there were four primary failure modes of the test specimens, namely, bending of the outermost solid wood panel on the tension side, splitting, a combination of bending and splitting of the outermost solid wood plate layer on the tension side, and splitting on the outermost solid wood board layer on the compressed side. The failure modes, split length, and direction of deflection for the specimens at different positions were recorded during the test, and summarized results are shown in Table 5.

**Table 5.** Failure modes of specimens.

| Specimen Number | Destruction Form | Splitting Length/mm | Flexure Direction |
|---|---|---|---|
| ZYG54-0-70 | Bending of one of the vertical solid wood laminate layers on the drawing side. | | 1 |
| ZYL54-0-70 | Split at medial margin.<br>Lateral edge splitting.<br>One of the vertical, solid wood laminate layers bent and bent on the drawing side. | 400<br>100 | 2 |
| ZYZ54-0-70 | A split at the inside edge of one of the vertical solid laminate layers. | 180 | 2 |
| ZYL34-0-70 | One of the vertical, solid wood laminate layers bent and bent on the drawing side.<br>Split at medial margin.<br>Lateral edge splitting.<br>Left vertical striped solid wood laminate splitting. | 420<br>180<br>240 | 3 |
| ZYL74-0-70 | | | 3 |
| ZYL58-0-70 | One of the vertical, solid wood laminate layers bent and bent on the drawing side.<br>Split at medial margin.<br>Lateral edge splitting.<br>Split on the inside edge of another vertical solid board layer on the pull side.<br>Lateral edge splitting. | 80<br>210<br>420<br>300 | 3 |
| ZYG54-30-80 | Splitting of one of the vertical solid wood slabs on the compressed side | 400 | 2 |
| DZYL54-0-70 | The bending of one of the vertical solid wood laminate layers on the drawing side. | | 1 |

Note: flexure direction "1" means nail-free side flexure, "2" means nail-free side flexure, "3" means nail-free lateral flexure, and "3" means nail-free lateral flexure with increasing load, The flexure direction is changed from the nailed side to the nail-less side.

### 3.3. Load-Deflection Curve of NCLT Panels

Test specimens exhibited large deflection under axial load. To investigate the effect of the number of nails in the overlap area on the mid-span displacement of the test specimen during the loading process, the load-deflection curve was recorded, as shown in Figure 5.

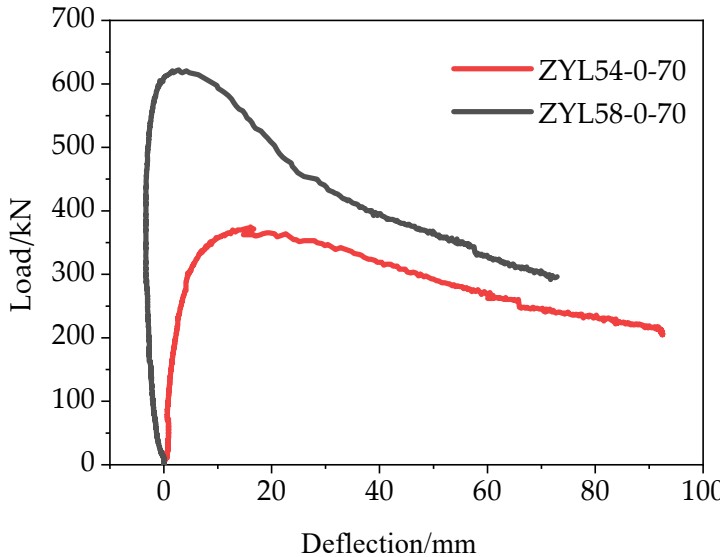

**Figure 5.** Load-deformation curves of stapling CLT panels with different number of nails.

The change of the load-deflection curve was recorded, and the influence of the number of nails in the overlap area on the yield and ultimate displacements of the specimen was analyzed. The yield and ultimate displacements of the test specimen are shown in Table 6. From the load-deflection curve, it can be observed that there were three stages from the beginning of the loading to failure: (1) the elastic stage, (2) the yield stage, and (3) the failure stage. The elastic phase extended before loading up to 80% of the ultimate load, where the test specimen behaved as elastic and the load versus displacement basically showed a linear relationship. In the yield stage, as the load increased the curve became less steep, indicating yielding. The specimen began to incur more obvious and rapid deflection and deformation. In the last failure stage, after reaching the load limit, the deflection increased more rapidly, and the curve began to drop as the load continued to increase. After decaying to about 80% of the limit load, the deflection increased sharply, and the specimen failed.

**Table 6.** Yield and ultimate displacement of stapling CLT panels with different numbers of nails.

| Number | Number of Nails | $P_u$/kN | $\Delta_m$ (mm) | $\Delta_y$ (mm) |
| --- | --- | --- | --- | --- |
| ZYL54-0-70 | 4 | 374.5 | 16.06 | 4.34 |
| ZYL58-0-70 | 8 | 622.1 | 2.77 | 2.99 |

Note: $P_u$ is the ultimate load, $\Delta_m$ is the corresponding deflection to $P_u$, $\Delta_y$ is the yield deflection, and the deflection is taken when the load reaches $0.8P_u$.

Figure 5 and Table 6 show that both the yield and ultimate displacements of the test specimen ZYL54-0-70 were more significant than that of specimen ZYL58-0-70. The yield and ultimate displacements were small since the bending direction of specimen ZYL58-0-70 changed during the loading process. The test specimen ZYL58-0-70 incurred slower development of deformation than that of specimen ZYL54-0-70 throughout the entire loading process. The test results show that increasing the number of nails in the overlap area reduced both the yield and ultimate displacements and slowed down the deformation rate of the test specimen.

Test specimens exhibited large deflection under axial pressure. To investigate the effect of the nail type on the mid-span displacement of the test specimen during the loading process, a load-deflection curve was developed based on the data collected in the test (Figure 6). The yield and ultimate displacements are given in Table 7. It can be observed that there were three stages including: (1) the elastic stage, (2) the yield stage, and (3) the failure stage. In the elastic stage, the deflection curves of specimens ZYG54-0-70 and ZYL54-0-70 basically coincided. As the load increased, the flexural deformation increased, but the flexural deformation did not increase significantly. The curve for specimen ZYZ54-0-70 was essentially a vertical straight line.

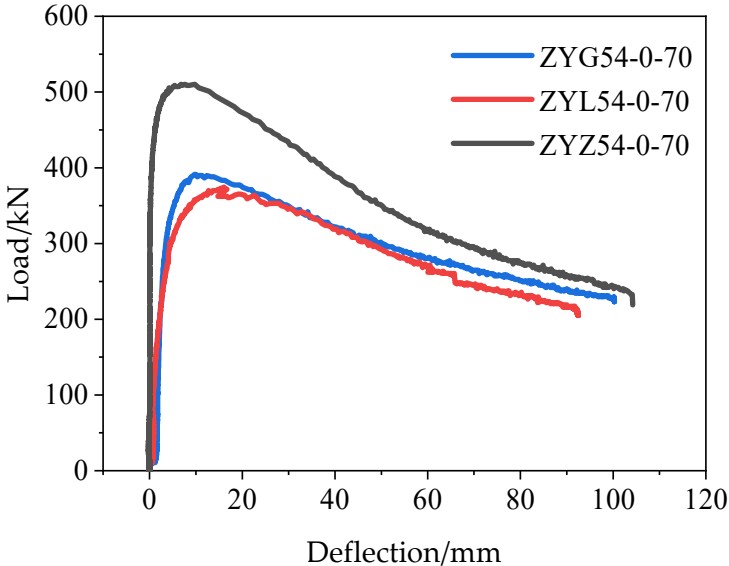

**Figure 6.** Load-deformation curves of stapling CLT panels with different types of nails.

**Table 7.** Yield and ultimate displacement of stapling CLT panels with different types of nails.

| Number | Type of Nail | $P_u$/kN | $\Delta_m$ (mm) | $\Delta_y$ (mm) |
|---|---|---|---|---|
| ZYG54-0-70 | Round nail | 391.2 | 10.00 | 3.82 |
| ZYL54-0-70 | Screw nail | 374.5 | 16.06 | 4.34 |
| ZYZ54-0-70 | Self-tapping screw | 530.1 | 9.77 | 0.56 |

Note: $P_u$ is the ultimate load, $\Delta_m$ is the corresponding deflection to $P_u$, $\Delta_y$ is the yield deflection, and the deflection is taken when the load reaches $0.8P_u$.

With increasing load, the test specimen did not undergo deflection or deformation, and the slope was larger compared to that of the other two specimens. In the yield stage, the curve for specimen ZYL54-0-70 was the smoothest, and the deflection deformation increased the fastest. The curve for specimen ZYZ54-0-70 was the steepest with slow deformation development. In the last failure stage, the rate of deflection growth of test specimens ZYG54-0-70 and ZYL54-0-70 was similar, while specimen ZYZ54-0-70 had the slowest deformation development compared with that of the other specimens.

Figure 6 and Table 7 show that among the three test specimens, ZYL54-0-70 had the highest yield and ultimate displacements, followed by ZYG54-0-70. The difference between the yield displacement and ultimate displacement, however, was relatively small, which shows that either using threaded nails or light round nails had little effect on the yield displacement and deformation rate of the test specimens. The use of threaded nails improved the maximum displacement of test specimens. Specimen ZYZ54-0-70 had the smallest yield and ultimate displacements since self-tapping screws effectively suppressed the development of cracks and reduced the rate of deformation.

Figure 7 shows the load deformation curves of stapling CTL panels with different angles of nails. The influence of the nail angle on the yield and ultimate displacements of

the specimen was analyzed. The yield and ultimate displacements of the test specimens are provided in Table 8.

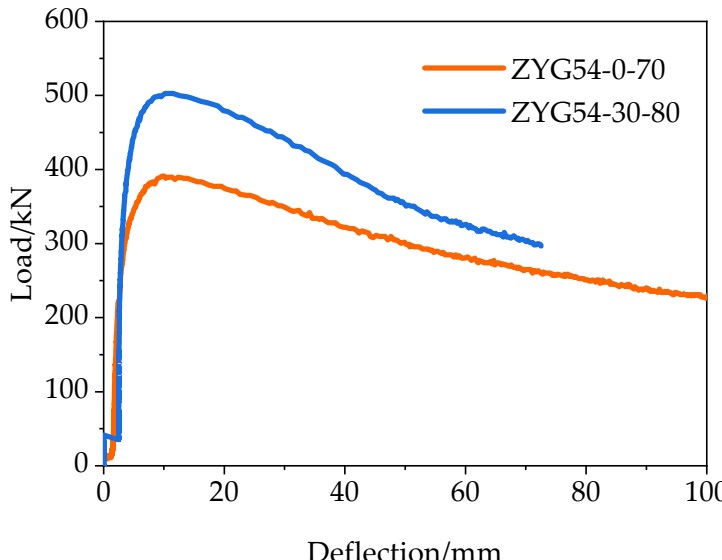

**Figure 7.** Load-deformation curves of stapling CLT panels with different angles of nails.

**Table 8.** Yield and ultimate displacement of stapling CLT panels with different angle of nails.

| Number | Nail Angle | $P_u$/kN | $\Delta_m$ (mm) | $\Delta_y$ (mm) |
|---|---|---|---|---|
| ZYG54-0-70 | 0 | 391.2 | 10.00 | 3.82 |
| ZYG58-30-80 | 30 | 502.8 | 10.15 | 4.12 |

Note: $P_u$ is the ultimate load, $\Delta_m$ is the corresponding deflection to $P_u$, $\Delta_y$ is the yield deflection, and the deflection is taken when the load reaches 0.8$P$.

Figure 7 indicates three stages from the start of loading to failure, namely, the elastic stage, yield stage, and failure stage. In the elastic stage, the curves of specimens ZYG54-0-70 and ZYG54-30-80 were nearly identical. As the load increased, the flexural deformation increased accordingly. However, the flexural deformation showed only a slight increase. In the second yield stage, the curve of specimen ZYG54-0-70 was smoother than that of specimen ZYG54-30-80, and the deformation developed faster, but the difference between the two was rather small. In the final failure stage, the curvature of the curve for specimen ZYG54-30-80 was small, indicating that specimen ZYG54-30-80 experienced slower deformation development with loading.

As can be observed in Figure 7 and Table 8, the yield and ultimate displacements for specimen ZYG54-30-80 were slightly larger than that of specimen ZYG54-0-70. Despite the fact that specimen ZYG54-30-80 had similar load and deflection changes in the elastic phase to that of specimen ZYG54-0-70, the deformation of specimen ZYG54-30-80 developed faster than its counterpart in the yield and failure phases. This implies that the test specimens were connected with the same type and same number of nails, which enabled nails to be at an angle on the wood surface, thereby slowing down the deformation rate of specimens and significantly improving the plastic deformation ability.

Figure 8 shows the load-deformation curves of stapling CLT panels with different slenderness ratios. By changing the height of the test specimen and the number of layers of the test plate, the effect of the slenderness ratio on the yield and ultimate displacements of the test specimen could be analyzed. The yield and ultimate displacements of the test specimens are shown in Table 9.

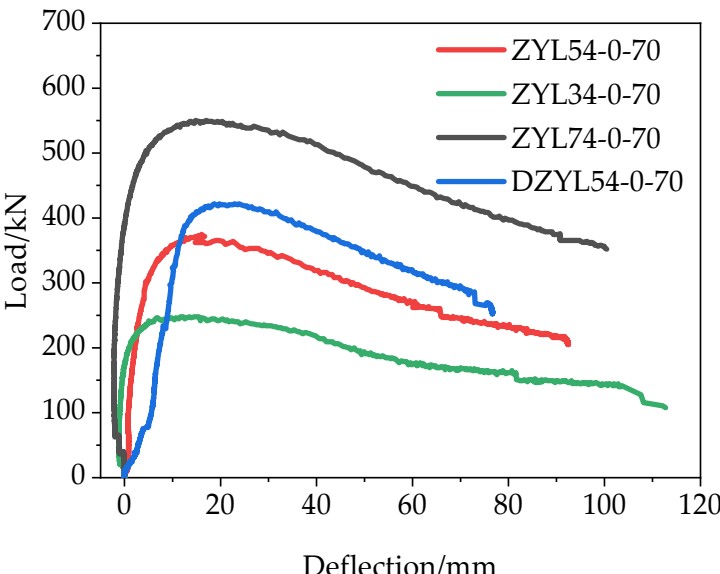

**Figure 8.** Load-deformation curves of stapling CLT panels with different slenderness ratio.

**Table 9.** Yield and ultimate displacement of stapling CLT panels with different slenderness ratio.

| Number | $\lambda$ | $P_u$/kN | $\Delta_m$ (mm) | $\Delta_y$ (mm) |
|---|---|---|---|---|
| ZYL34-0-70 | 67.11 | 247.7 | 11.64 | 0.85 |
| ZYL54-0-70 | 40.47 | 374.5 | 16.06 | 4.34 |
| DZYL54-0-70 | 32.80 | 421.8 | 18.73 | 10.75 |
| ZYL74-0-70 | 28.71 | 567.9 | 17.01 | 1.27 |

Note: $\lambda$ is for aspect ratio, calculation based on full section moment of inertia; $P_u$ is the ultimate load, $\Delta_m$ is the corresponding deflection to $P_u$, $\Delta_y$ is the yield deflection, and the deflection is taken when the load reaches $0.8P_u$.

From the load-deflection curve, it can be noted that there were again three stages from start to failure. In the first elastic stage of loading, the deflections of the test specimens ZYL74-0-70 and ZYL34-0-70 were negative. The initial deflection of the two specimens occurred from the nail-less side to the nailed side, and deformation was not obvious except that the deflection deformation increased with the load. When the load reached about 200 kN, the direction of deflection changed, and deflection increased with loading. The specimen DZYL54-0-70 had large deformation due to processing errors at the initial stage of loading. Therefore, the deformation was large. When the load reached around 100 kN, the rate of deformation decreased slightly with increase of the load. Deformation for the test specimen ZYL54-0-70 increased linearly with the increase of load. Within the normal deformation range of the four specimens, changes in rate of deformation were comparable. In the second yield stage (Figure 8), the curve of the four specimens was comparable, that is, the increase in deflection rate was basically similar as the load increased. In the terminal failure stage, the deflection rate for specimens ZYL34-0-70, ZYL54-0-70, and ZYL74-0-70 was similar. After the test specimen DZYL54-0-70 entered the damage stage, the deformation developed faster, and the specimen quickly lost its load bearing capacity.

### 3.4. Load-Strain Curve of NCLT Panels

According to the load-strain curve (Figure 9), corresponding to the load-deflection curve (Figure 5), the specimen experienced three stages from the start of loading to failure: elastic stage, yield stage, and failure stage. During the entire loading process, the rate of tensile strain increase for the two specimens at the yield and failure stages were greater than the increase rate of the compression strain, reflecting a similar curve change trend. In addition, the change in compression strain at the two sides of the test specimen ZYL58-0–70 in the elastic stage was comparable. This indicates that the number of nails in the overlap area had little effect on the change of both tension and compression strains on both sides of

the specimen. When the test specimen failed, only slight folds appeared on the compressed side of the plate.

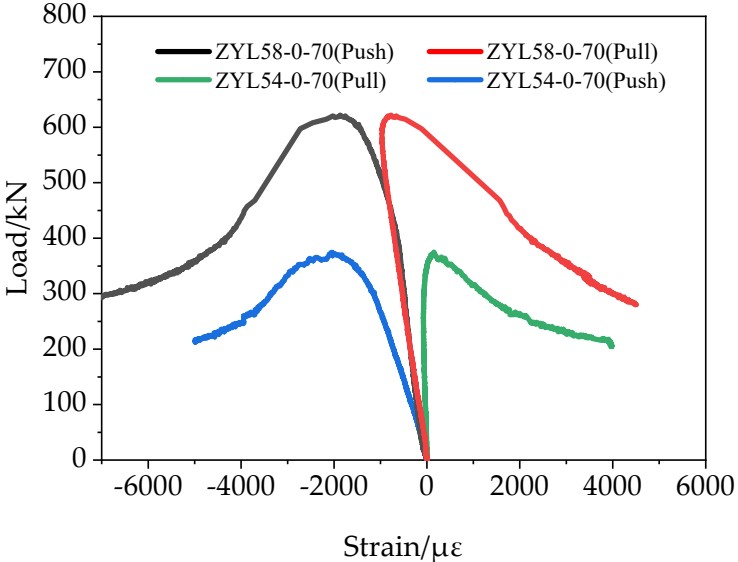

**Figure 9.** Load-strain curves of stapling NCLT panels with different numbers of nails.

The relationship between specimen load and midspan strain and the effect of nail type on midspan strain is illustrated in Figure 10.

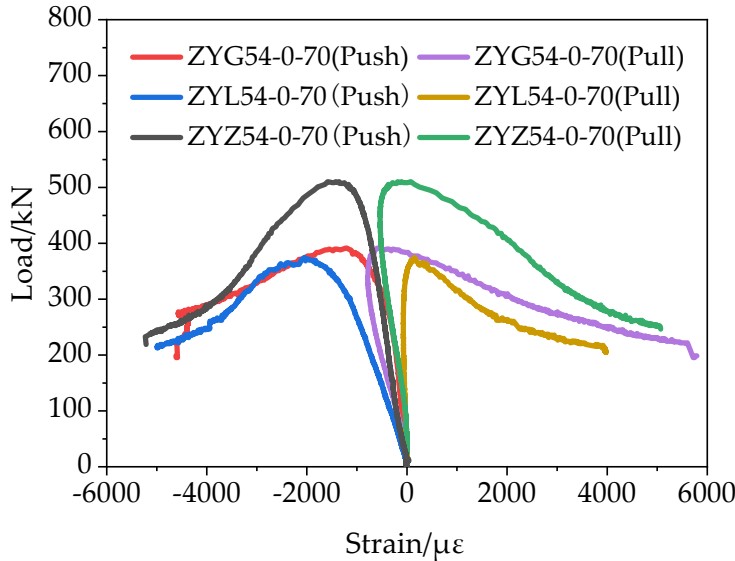

**Figure 10.** Load-strain curves of stapling CLT panels with different types of nails.

During the entire loading process, the test panels connected by self-tapping screws exhibited the slowest growth in tension and compression strain compared to that of the other two specimens. The strain value on the compression side was consistently larger than that at the tension side in specimens ZYG54-0-70 and ZYL54-0-70. When all specimens failed, only slight creasing occurred on the compression side of the plate. The tension side of the plate, however, showed cracking, bending, and splitting, indicating that the compressive strength parallel to the grain was much higher than that of the tensile strength. The strain values of the two sides of specimen ZYZ54-0-70 were comparable. This can be attributed to the effect of self-tapping screws which delayed the development of cracking, thus dramatically enhancing the tensile strength of the vertical grain solid wood slab.

The relationship between specimen load and mid-span strain and the effect of the nail angle on the mid-span strain when nailing into the wood surface were also investigated (Figure 11). The rate of strain change for the tension side was less than that of the compression side for specimen ZYG54-30-80 at all load stages. The tensile side of specimen ZYG54-0-70 was under compression during the elastic and yielding phases. The rate of strain increase on both tension and compression sides during the failure phase was similar. When the test specimen was broken, only slight creases appeared on the compressed side of the test plate. In addition, one plate on the tension side of specimen ZYG54-0-70 was bent. In contrast, splitting of one plate layer occurred on the tension side of specimen ZYG54-30-80. This indicates that the inclination of the nail and the wood surface at a certain angle can slow down the increasing rate of tensile strain and improve the tensile strength of the layer.

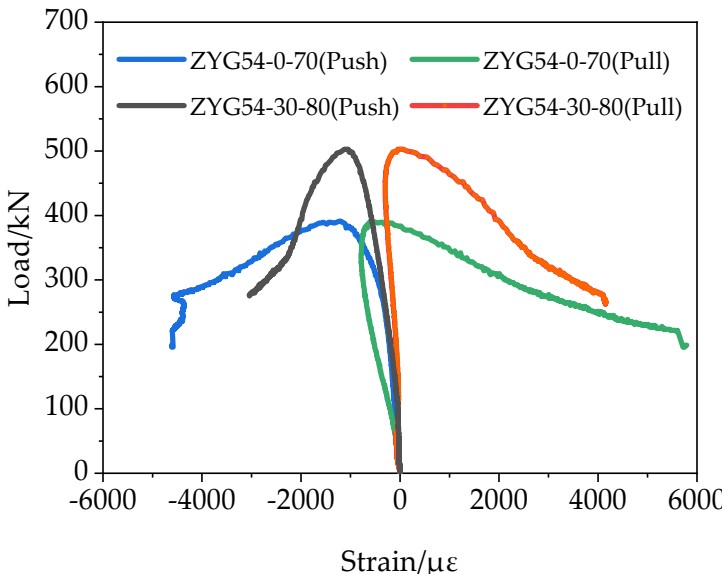

**Figure 11.** Load-strain curves of stapling CLT panels with different angle of nails.

Figure 12 illustrates the relationship between the specimen load and mid-span strain, along with the effect of the slenderness ratio on mid-span strain. During the entire loading process, the test specimens ZYL54-0-70, DZYL54-0-70, and ZYL74-0-70 exhibited similar rate of increase in tensile and compressive strain during the yield and the failure stages, i.e., when the slenderness ratio was less than 42, it had a minor effect on the rate of change of tensile and compressive strains during the yield and failure stages. The compression side of specimen ZYL34-0-70 was rather under tension during the elastic phase and yielding phase. The increasing rate of strain on both tension and compression sides during the failure phase were comparable. At specimen failure, only slight folds appeared on the compressed side plate. Plate layers of the tension side of specimen ZYL34-0-70 were bent and split. Splitting also occurred in the middle plate layer. Specimen ZYL34-0-70 was the most severely damaged among the four test specimens. One of the slabs on the tension side of specimen ZYL54-0-70 was bent and split, but the length of the split was small. Bending was only found in specimen DZYL54-0-70. In contrast, the deformation of specimen ZYL74-0-70 was very small, while only minor cracks appeared on the surfaces of both tension and compression sides with no bending and splitting. This implies that reducing the slenderness ratio of the specimen can reduce the rate of tensile strain on the tensile side of the specimen, improve the stripe tensile strength of the slab, and reduce damage on the tensile side.

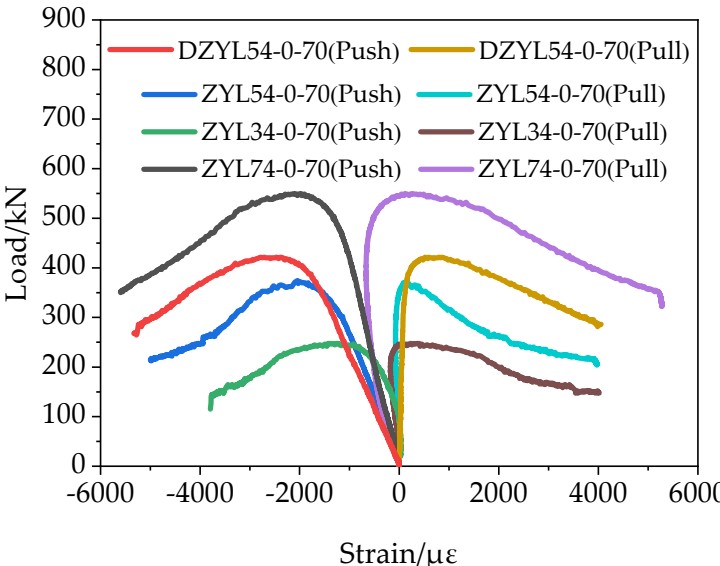

**Figure 12.** Load-strain curves of stapling CLT panels with different slenderness ratio.

### 3.5. Axial-Load Bearing Capacity

To adequately select the quantity of material and assess the safety of an NCTL structure, it is necessary to carry out a calculation of the bearing capacity of NCLT panels. The combination beam theory based on mechanical connection is the most commonly used analytical method in Europe. Therefore, the calculation formula for the effective bending stiffness and effective section moment of inertia of the NCLT panels was derived as per Appendix B of the European Specification of Timber Structures (Equations (1) and (2)). The components shall conform to the following assumptions: the shear deformation of the longitudinal deck is negligible; only the rolling shear deformation of the transverse ply is considered. The combination coefficient is introduced, and the equation to calculate the combination coefficient is:

$$EI_{eff} = \sum_{i=1}^{n} E_i I_i + r_i E_i \alpha_i^2 \tag{1}$$

$$I_{eff} = \frac{EI_{eff}}{E} \tag{2}$$

$$r_i = \frac{1}{1 + \left( \pi^2 \frac{E_i A_i}{l^2} \right) \left( \frac{h_i}{G_R b} \right)} \tag{3}$$

where $E_i$ is the modulus of elasticity of layer $i$ (MPa), $I_i$ is the moment of inertia of the $i$-th layer (mm$^4$), $\alpha_i$ is the distance between the center of the $i$-th layer and the neutral axis of the NCLT panels (mm), $r_i$ is the combination coefficient of layer $i$, $l$ is the free length of component along length direction (mm), $h_i$ is the thickness of transverse plate (mm), $B$ is the breadth of section (mm), and $G_R$ is the shear modulus of transverse laminate (MPa).

The bearing capacity of axial compression members is checked according to stability, and the stability coefficient is calculated as shown in Equations (4)–(7). When the intensity level of tree species is TC11, TB17, TB15, TB13 and TB11, the stability coefficient is calculated as follows:

$$\phi = \frac{1}{1 + \left( \frac{\lambda}{65} \right)^2} \qquad \lambda \leq 91 \tag{4}$$

$$\phi = \frac{2800}{\lambda^2} \qquad \lambda > 91 \tag{5}$$

When the strength grade of tree species is TC17, TC15, and TB20, the stability coefficient is calculated according to Equations (6) and (7).

$$\phi = \frac{1}{1 + \left(\frac{\lambda}{80}\right)^2} \qquad \lambda \leq 75 \tag{6}$$

$$\phi = \frac{3000}{\lambda^2} \qquad \lambda > 75 \tag{7}$$

where $\phi$ is the stability coefficient of axial compression members, $\lambda$ is the slenderness ratio of components, $\lambda = \frac{l_0}{i}$, $i$ is the radius of rotation of component section (mm), $i = \sqrt{\frac{I}{A}}$, $l_0$ is the calculated length of the component (mm), $I$ is the full-section moment of inertia of the component (mm$^4$), $A$ is the full cross-section area of members (mm$^4$).

Using the Chinese code for design of wood structures and the calculation method of axial compression of wood members in Canada, the theoretical value of bearing capacity of NCLT panels is obtained, which is compared with the experimental ultimate bearing capacity, as shown in Tables 10 and 11, to select the calculation method suitable for the axial compression bearing capacity of NCLT panels. Results show that the ratio of F to N2 is the closest to 1.00 in specimens ZYL54-0-70, DZYL54-0-70, ZYL74-0-70, ZYG54-0-70. F is the bearing capacity of axial compression members under stable control (N) and N is the theoretical ultimate bearing capacity based on section moment of inertia calculated by different theories according to China construction specifications. This implies that according to the stability checking formula of axial compression members in the GB50005-2012 Code for the Design of Wood Structures, the axial compression bearing capacity of NCLT panels is calculated using the effective section moment of inertia obtained via the composite beam theory. The theoretical values are in good agreement with the corresponding experimentally measured data. In specimens ZYL34-0-70, ZYZ54-0-70, ZYL58-0-70, ZYG54-30-80, the F/N1 ratio is slightly greater than 1.00. Therefore, according to the stability checking formula of axial compression members in the GB50005-2012 Code, the axial compression bearing capacity of NCLT panels is calculated using the full cross-section moment of inertia, and the theoretical values were in good agreement with the corresponding experimental values.

**Table 10.** Comparison between experimental and theoretical values based on domestic and foreign criteria.

| Test Specimens No. | Trial Value F/kN | N1/kN | N2/kN | N3/kN | N4/kN | N5/kN | N6/kN | N7/kN | N8/kN |
|---|---|---|---|---|---|---|---|---|---|
| ZYL34-0-70 | 247.7 | 203.1 | 172.93 | 105.02 | 138.16 | 266.02 | 278.36 | 182.04 | 234.62 |
| ZYL54-0-70 | 374.5 | 503.8 | 392.13 | 363.64 | 473.47 | 542.94 | 528.06 | 514.83 | 558.60 |
| DZYL54-0-70 | 421.8 | 557.2 | 431.29 | 435.30 | 530.04 | 577.40 | 558.05 | 559.52 | 586.88 |
| ZYL74-0-70 | 567.9 | 819.2 | 632.15 | 734.43 | 809.24 | 767.16 | 742.26 | 759.74 | 774.51 |
| ZYG54-0-70 | 391.2 | 503.8 | 392.13 | 363.64 | 473.47 | 542.94 | 528.06 | 514.83 | 558.60 |
| ZYZ54-0-70 | 530.1 | 503.8 | 392.13 | 363.64 | 473.47 | 542.94 | 528.06 | 514.83 | 558.60 |
| ZYL58-0-70 | 622.1 | 503.8 | 392.13 | 363.64 | 473.47 | 542.94 | 528.06 | 514.83 | 558.60 |
| ZYG54-30-80 | 502.8 | 503.8 | 392.13 | 363.64 | 473.47 | 542.94 | 528.06 | 514.83 | 558.60 |

Note: *N1, N2, N3,* and *N4* are calculated following domestic norms, *t*he effective cross-sectional moment of inertia is calculated from the combined beam theory, the compound theory and the shear theory, respectively. *N5, N6, N7, N8* are calculated according to Canadian specifications, according to the composite beam theory, the composite theory and the shear theory, the effective slenderness ratio is calculated by using slenderness ratio.

Given defects that occur in wood, such as decay, as well as common processing errors, the design of NCLT panels shall be relatively conservative. At the same time, the production cost shall be reduced, and the performance shall be safe and reliable. Therefore, it is recommended to multiply a safety factor of 0.9 for design code purposes. The calculation formula becomes:

$$F = 0.9\phi f_c AF = 0.9\phi f_c A \tag{8}$$

where $F$ is the bearing capacity of axial compression members under stable control (N), $f_c$ is the design value of compressive strength of wood along the grain (N/mm$^2$), $\phi$ is the stability coefficient of axial compression members, and $A$ is the full cross-section area of members (mm$^2$).

Among these, the stability coefficient is according to Equations (4)–(7), but in the process of the slenderness ratio calculation, it is necessary to consider the influence of the nail type, nail number, nail angle, and plate layer number on the moment of inertia of NCLT panels as follows: (1) when the number of the platelayer is three with self-tapping screws, and eight nails in the overlapping area, or the nail is inclined at an angle with the surface of the wood, the full-section moment of inertia calculation can be adopted; and (2) when the number of layers is five or seven, and the nail type is a threaded nail or light circular nail, the effective section inertia moment obtained according to the composite beam theory may be calculated according to Equation (1).

**Table 11.** Comparison with ratio of experimental and theoretical values.

| Test Specimens No. | F/N1 | F/N2 | F/N3 | F/N4 | F/N5 | F/N6 | F/N7 | F/N8 |
|---|---|---|---|---|---|---|---|---|
| ZYL34-0-70 | 1.22 | 1.43 | 2.36 | 1.79 | 0.93 | 0.89 | 1.36 | 1.06 |
| ZYL54-0-70 | 0.74 | 0.96 | 1.03 | 0.79 | 0.69 | 0.71 | 0.73 | 0.67 |
| DZYL54-0-70 | 0.76 | 0.98 | 0.97 | 0.80 | 0.73 | 0.76 | 0.75 | 0.72 |
| ZYL74-0-70 | 0.69 | 0.92 | 0.77 | 0.70 | 0.74 | 0.77 | 0.75 | 0.73 |
| ZYG54-0-70 | 0.78 | 1.00 | 1.08 | 0.83 | 0.72 | 0.74 | 0.76 | 0.70 |
| ZYZ54-0-70 | 1.05 | 1.35 | 1.46 | 1.12 | 0.98 | 1.00 | 1.03 | 0.95 |
| ZYL58-0-70 | 1.23 | 1.59 | 1.71 | 1.31 | 1.15 | 1.18 | 1.21 | 1.11 |
| ZYG54-30-80 | 1.00 | 1.28 | 1.38 | 1.06 | 0.93 | 0.95 | 0.98 | 0.90 |

## 4. Conclusions

This study investigated the structural behaviour of NCLT panels as a more resilient alternative to conventional cross-laminated timber. The effects of various design parameters on the axial compression behavior of NCLT panels were explored. The failure modes of test specimens and deformation of nails were analyzed, and the effects of four key design parameters, including the nail type, nail angle, slenderness ratio, and the number of nails in the superimposed area on the displacement, strain, ultimate bearing capacity, yield-bearing capacity, ductile performance, and elastic recovery ability of the span were thoroughly investigated. Based on experimental results and observations, the following conclusions can be drawn:

- Compared with light round nails and thread nails, the specimen connected by tapping screws attained the best compressive performance.
- The increase in the number of nails in the superimposed area greatly reduced both the yield and ultimate displacements of the specimen in the middle span, but had little effect on the rate of increase of tension and compression strain on both sides of the specimen.
- Under drawing action, the pullout anchoring effect of the plate connected by tapping screws was best compared with that using light round nails and thread nails.
- The anti-drawing capability of the entire test specimens can be improved considerably i by increasing the number of nails in the overlapping area and inclining the nails to the wood surface.
- The compressive bearing capacity of the specimen increased with increasing the number of nails in the superimposed area and the tilt angle between the nail and the wood surface. However, the ductility and ability of elastic recovery of the specimens decreased in both cases.
- A certain angle between the nail and wood surface reduced the tensile strain on the tensile side of the specimen and had little effect on the yield and ultimate displacements at the middle of the span.

- The smaller the slenderness ratio, the larger the midspan ultimate displacement, and the slower was the tensile strain increase, the higher the compressive bearing capacity.
- Despite of advantages of NCLT panels, they still have some shortcomings, such as the potential corrosion of nails, that need to be addressed in future concerted research.

**Author Contributions:** Conceptualization, Y.Z. and M.L.N.; methodology, Y.Z. and M.L.N.; validation, Y.Z., M.L.N.; formal analysis, Y.Z., X.G., M.L.N.; investigation, X.G.; resources, Y.Z.; data curation, X.G., L.V.Z. and A.R.S.; writing—original draft preparation, X.G., Y.Z., M.L.N.; writing—review and editing, M.L.N., L.V.Z. and A.R.S.; visualization, X.G., Y.Z., L.V.Z.; supervision, Y.Z., M.L.N.; project administration, M.L.N.; funding acquisition, Y.Z. All authors have read and agreed to the published version of the manuscript.

**Funding:** This research project was funded by the China Liaoning Province Major Special Program (2020020307-JH1/103-02-02).

**Institutional Review Board Statement:** Not applicable.

**Informed Consent Statement:** Not applicable.

**Data Availability Statement:** All data is provided in the manuscript.

**Conflicts of Interest:** The authors declare no conflict of interest.

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
