# Peer review of "Experimental Investigation on Axial Compression of Resilient Nail-Cross-Laminated Timber Panels"

_sustainability, doi:10.3390/su132011257_

Round 1
Reviewer 1 Report
dear Authors
The topic is interesting and the paper is well-written. Anyhow some revisions are needed as follows:
line 46-51. you forgot also acoustics . In addition please add for every characteristics (e.g. fire protection, energy saving etc) one reference each
some hints https://doi.org/10.1016/j.jobe.2021.103066 and 10.2749/101686608784218716 and 10.1016/j.firesaf.2017.05.002 etc
section 2.1 a picture or a graphical scheme would help the reader to understand better
Section 2.2 a scheme of the strain gauges placement would definitely help following the description
line 122 spruce-pine-fir Why those three woods? why not other? please explain
the first section of the 3.5 section is to be moved in the "materials and method" section
Author Response
Thank you for the insightful review and constructive criticism, which enhanced the quality of the revised manuscript. Please find attached a point-by-point response explaining how we addressed each of your comments and the corresponding changes in the revised manuscript.

Reviewer 2 Report
This paper reports the results of a research on the structural performance of Nail-Cross-Laminated Timber (NCLT) under axial compression loads. I would like to see some revisions on the paper before making any recommendation for its publication. Please look at my other comments below.
General comments:
The authors claim that NCLT is novel, but there are similar products developed by other researchers with a more or less similar approach (e.g., see, for example, Hosseinzadeh et al. 2020 at https://doi.org/10.1080/17480272.2020.1800089). NCLT has some shortcomings that need to be addressed as well, for example, the use of nails or screws instead of glue, makes it difficult to cut the panels after its production as the saw has to cut through the steel nails/screws! What are the authors suggestions to deal with this kind of difficulties in practical use? Additionally, nails/screws are susceptible to corrosion and this can reduce their performance over time; what are the authors plans to handle this issue?
More specific comments:
The introduction is focused on the introduction of conventional CLT, its applications, its benefits, and its shortcomings. I was surprised that the authors have made no mention of NCLT that has been developed in the previous research works, even though some of the authors themselves have publications on this subject (e.g., https://doi.org/10.3390/app10175983)! There is also no mention of any study on nail-laminated timber (NLT), which is quite relevant to the scope of this research. Therefore, the main literature has not been reviewed/cited properly. I suggest that the authors make a detail analysis of the available literature on both NLT and NCLT and describe them in two or more paragraphs in the introduction section.
On line 54, the authors claim “…relatively low service life of the adhesive connection…”. Please provide a reference for this claim.
On line 55, also provide a reference for this claim: “…poor connection exhibiting common glue joint cracking and plate layer separation…”.
Section 2.1: Please include the name of the materials of the nails and screws as well as the structural grade of the timber used.
Table 3: where do these numbers come from? Did you experimentally determine the material properties? What standard was used for this purpose?
In the conclusion section, the authors claim: “NCLT panels have exceptional advantages over conventional CLT panels, including durability, eco-friendliness, high yield strength, and better insulation”. However, in this paper you have only evaluated the structural response under compression load; how do you conclude based on a compression test that your NCLT is more durable and ecofriendly than the conventional CLT? How can you conclude based on a compression test that your NCLT is a better insulator than conventional CLT? If these claims are made, the authors must include data in the paper to support them.
In the conclusion section, the authors should try to only include remarks relevant to their conducted experiments/analysis PLUS a paragraph pointing out to the limitations/shortcomings of NCLT that I have mentioned above.
Author Response

(The authors gave the same response as above.)

Reviewer 3 Report
This study investigates the structural behavior of Nail-Cross-Laminated Timber panels as a more resilient alternative to conventional cross-laminated timber. Further literature review should be strongly considered to better place this in the broader context of timber engineering, please consider expanding this.
10.1007%2Fs00107-015-0999-5
10.1016/j.istruc.2020.03.034
10.1016/j.istruc.2020.07.072
Author Response

(The authors gave the same response as above.)

Reviewer 4 Report
The presented paper is very interesting, because CLT panels are now a very important construction material.
The paper needs some improvement, because now it has many understatements:
- Literature review is very short and not full – there is some literature on scientific investigation of CLT mechanical parameters - please add.
- The description of tested panels is not complete: please add info on: the storage conditions, wood humidity and please answer the question: are you sure that CLT is an isotropic material?? As it is known it is orthotropic, so the directions of mechanical properties and the way of cutting must be given. It is also a question for equations given in chapter 3.5 – they do not consider the material orthotropy – why?
The figures showing the distribution of nails and displacement gauges must be presented. Also some schematic figure showing the nail angle.
How many samples of each kind were tested, what was the repeatability of the results (any statistics were made)?
What was the conditions during testing (temperature, air humidity, were they always the same?
- What do you mean by “full-scale” for the samples?
- All figures showing resultant curves are low quality and must be improved.
- Please comment what is the meaning of negative values of strain, e.g. in Fig. 9.
- Conclusions should be improved: why there is a difference in strength behavior when using different connectors/nails? Any physical explanation?
Some editorial corrections:
- In table 3 please change Mpa into MPa.
- In table 3 and the rest of the test please change the notification of measured value and the unit. For example in table 3 there is Dm/mm, what can be understood as increase of meters per millimeter. Please change those notifications into Dm (mm).
Author Response

(The authors gave the same response as above.)

Round 2
Reviewer 2 Report
I would like to thank the authors for considering the issues.
The comments have been addressed properly. However, the introduction can still be improved, specifically the main literature in the area of NLT is still missing here.
On line 63: Reference number 24 is not about NLT, but it is more related to nail joints in general. Try to cite more relevant publications to NLT. Here are two main sources that you can use:
https://open.library.ubc.ca/soa/cIRcle/collections/ubctheses/24/items/1.0354482
https://link.springer.com/article/10.1007/s00107-019-01408-9
A minor typo:
On line 60: Replace NCLT in this sentence with NLT: ….”NCLT can be produced with any dimension”...
Author Response
The authors would like to thank the Reviewer for the valuable feedback and recommendations, which enhanced the overall quality of the manuscript. The new manuscript has addressed the comments raised by the Reviewer in detail as explained in the attached document. Additional pertinent literature has been analyzed as recommended and has been highlighted in the list of references.

Reviewer 4 Report
The authors answered all my questions and fully corrected the paper.
Author Response
The authors would like to thank the reviewer for the valuable feedback and recommendations which improved the final manuscricpt.